# The Decorative Auspicious Elements of Traditional Bai Architecture in Shaxi Ancient town, China

Hua Zhao [1], Zongsheng Huang [1,2,*], Caijie Deng [3] and Yuxin Ren [1]

1   College of Architecture and Urban Planning, Guizhou University, Guiyang 550025, China
2   College of construction engineering, The College of Humanities and Science of Guiyang, Guiyang 550025, China
3   College of Forestry, Guizhou University, Guiyang 550025, China
*   Correspondence: hzsxjh@126.com

**Abstract:** The lucky cultural characteristics of traditional architecture are of importance. It shows what makes a place unique and the spiritual and material goals people have there. It is thus vital to understand the lucky cultural characteristics of traditional villages. This paper attempts to explore the auspicious cultural attributes of the town. We are aiming to reveal the current status of cultural integration in Shaxi's ancient town so that we can find the problems arising from the development process. Moreover, the research subject is the lucky element of Shaxi Bai's traditional architecture. Lucky themes include lotuses, unicorns, phoenixes, etc. The research was qualitative and quantitative, so we begin by identifying the lucky elements. Then, we used methods for analysing diversity and complex networks to determine their diversity indices and network model indicators. These findings show the old town's diversified, lucky culture. However, Buddhist culture dominates the multicultural makeup. Furthermore, the native Bai culture is also conserved and preserved.

**Keywords:** lucky element diversity; lucky cultural characteristics; co-occurrence network; Shaxi ancient town; Bai traditional architecture; multicultural integration; indigenous culture

## 1. Introduction

Shaxi Ancient Town is situated in the Dali area of China, where the Bai people live. Today, it is the only typically well-preserved settlement. It combines Confucianism, Buddhism, and the commercial horse band culture. The early Ming and Qing architectural culture also influences the old town. It reflects the multicultural makeup of traditional Bai architecture. The ancient town of Shaxi has been home to one of the world's endangered buildings since 2002. Due to its modest village design and historical and cultural importance, Shaxi has become a renowned tourist attraction and social centre. Moreover, it has attracted tourists and researchers from all over the world. The Shaxi revitalisation project [1] has preserved the historic town in various ways. However, tourism and urbanisation have brought foreign cultures into the town. Many researchers are trying to explore this issue from different perspectives, as it is vital to find a new way to protect and bring life back to the traditional villages, such as cultural heritage [2], cultural tourism [3,4], and national culture [5]. Luck expresses the people's view of a brighter future. Different lucky cultures may reflect various folklore and cultural backgrounds. Common lucky elements include objects, behaviours, language, words, and numbers. Traditional architecture's lucky elements refer to the decorative motif in each building component. They also represent a desire for a better life. This study analyses the blessed cultural characteristics of traditional Shaxi Bai architecture.

Today, traditional village studies focus on cultural perspectives [6]. Ao [7] talked about how influential culture is. Moreover, America [8] explained how culture and architecture are linked. Xia [9] also said that local culture should be a part of historic districts. In the past few years, regions with distinct ethnic traditions have been the focus of Chinese

architecture research, such as the southern part of Fujian [10,11], the northern part of Shaanxi [12], and Jinzhong [13]. Only some studies about the Bai region in northwest Yunnan are helpful. Aside from that, many studies on lucky culture focus on traditional residential architecture. Furthermore, most researchers have examined what architectural decorations mean to different cultures. Similarly to Xuan Ran [14], he looked at the artistic and cultural values already there. Wang Yan [15] demonstrated the significance of decoration in contemporary architectural design. Du Y [16] emphasised how the method of lucky decorations could be applied in practice. It integrates traditional architectural components such as random decorations into modern building design. Furthermore, this helps the Chinese architectural market change and preserves fortunate culture. Hongyan Xiang [17] demonstrates lucky culture through traditional building decorations, which are auspicious. It examined how traditional architecture preserves and promotes good luck. In conclusion, most research describes random things qualitatively, focusing on identification and appraisal. Few quantitative research has examined lucky element diversity and co-occurrence network properties.

We need qualitative and quantitative methods to reveal traditional architecture's lucky cultural traits. This study focuses on Shaxi, an old town in the Bai region of Yunnan Province. The first step was to carry out field research to find lucky elements. Then, we figured out the diversity index of lucky elements by diversity analysis. Moreover, we worked out the co-occurrence network index by complex network analysis. The last step is to compare and examine the calculations. This will reveal the lucky cultural characteristics of the classic old-town architecture. Next, the multicultural-indigenous culture link in the ancient town is investigated. In the historic town, to exhibit the cultural perspective of the Bai people, Shaxi may be an example of conserving and developing other traditional villages.

## 2. Overview of the Study Area and Research Methods

### 2.1. Overview of the Study Area

Shaxi Ancient Town is situated in Dali, Yunnan Province, China (See Figure 1). In 2001, the WMF (World Monuments Foundation) identified Shaxi as "the only ancient market on the Tea Horse Trail that still survives". In 2002, it was included on the World Endangered Architectural Heritage List [18]. It is incorporated as "a Yunnan province famous historical and cultural town", "a nationally famous historical and cultural city", "a famous tourist town in Yunnan", "108 Chinese village business cards", and "the village directory of Chinese rural cultural heritage monuments". This study is based on the "Historic Town Plan of Shaxi" from 2005. The plan shows where the old town reserve is. The region of protection ($36.11°$ $11–26.19°$ N, $99.45°–99.58°$ E) has an area of 8.4 hm$^2$. Moreover, it is situated in the North Temperate Cool Layer at an average height of 2100 m. It has an average annual temperature of 12.3 °C, yearly sunshine hours of 2400 h, and annual precipitation of 790 mm [19]. The protected area centres on Xingjiao Temple, Sifang Street, and Ancient Theatre. South and North Guzong Lane are next to it, and traditional buildings are around it [20]. The share of conventional architecture is also relatively high. The population comprises Han, Bai, Yi, Lisu, Naxi, and other ethnic groups [21]. Bai accounts for 85 per cent of the entire population and is the largest ethnic group [22]. Bai is the fifteenth most significant ethnic minority in China. Thus, this region's traditional Bai architecture represents ancient Shaxi's lucky culture.

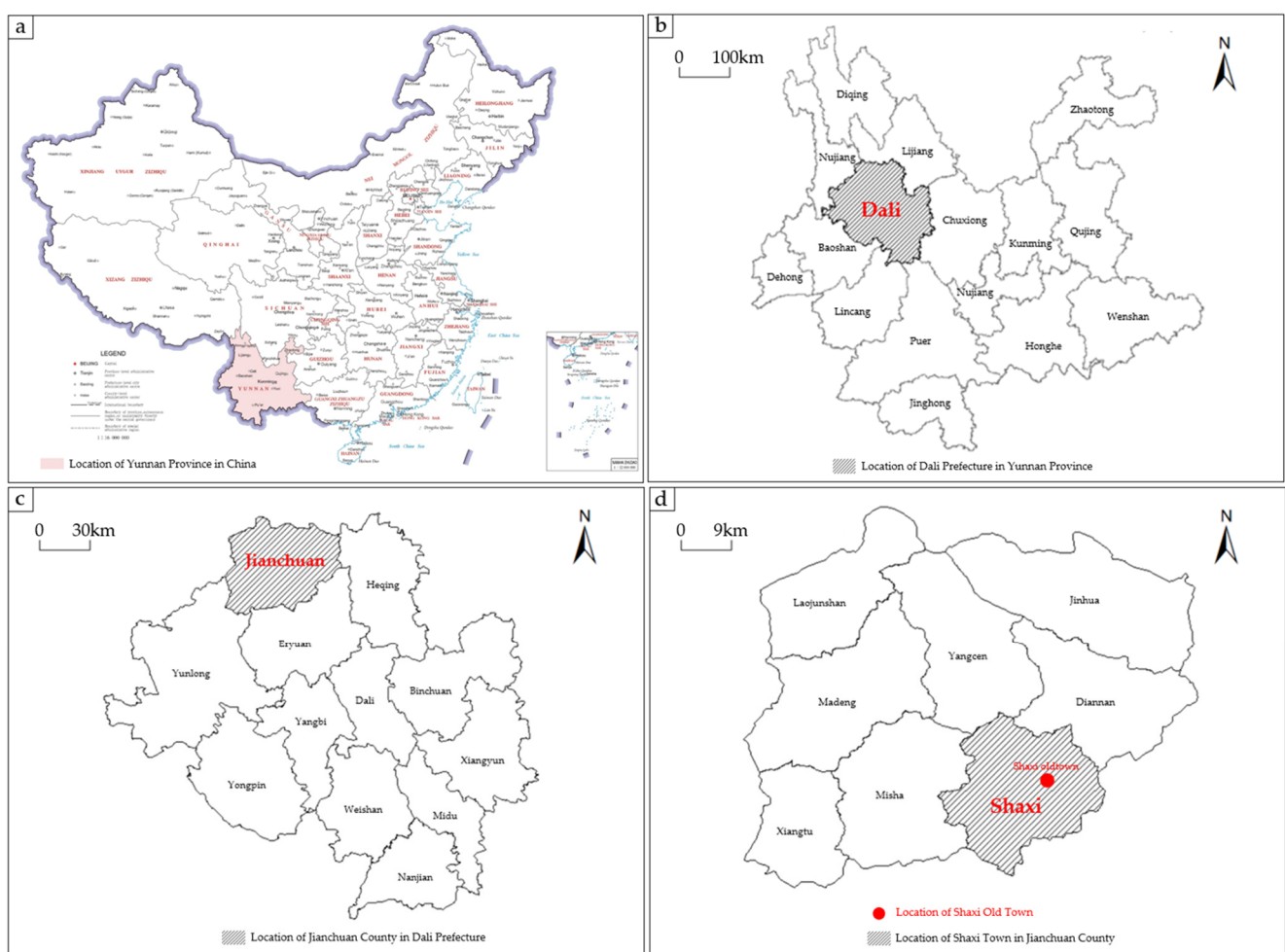

**Figure 1.** The geographical location of the study area. (**a**) Is from the Ministry of Natural Resources of China, and the review number is GS (2019) 1686. (**b**–**d**) are based on (**a**); the authors used ArcGIS software for self-mapping.

*2.2. Research Methodology*

2.2.1. Overall Idea

We begin by reviewing the literature and field research. We selected the historic town's protection area as our study location. Under the "Shaxi Historical and Cultural Town Conservation Plan", we also identified six old town building types by function. They are the Xingjiao Temple, the Patron God Monastery, the Ancient Theatre, the Essential Privileged Dwellings, the Generally Sheltered Dwellings, and the Stores. Second, we, local experts and craftsmen, discussed lucky on-site elements. They include the location (building components), symbolic significance, materials, and presentation methods. Then, we compiled a list of the lucky aspects found in each of the six buildings. Local experts are non-hereditary inheritors of traditional architecture and artisans have been building historic buildings for more than ten years. Finally, we calculate the auspicious element diversity index. It is to compare multicultural integration across architectural types. We also create a network model with propitious elements and get an index. It is to compare lucky culture compositions. Then, we compared and examined the calculations. It is to reveal the lucky cultural characteristics of classic old-town architecture. Next, the multicultural-indigenous culture link in the ancient town is investigated to exhibit the cultural perspective of the Bai people.

2.2.2. Determination of the Diversity of Auspicious Elements

In this study, each building type is considered a "community" and each auspicious element is considered a "species". During the field research, the Xingjiao Temple, the Patron God Monastery, and the ancient theatre were the only building groups in the old town. Therefore, only a single sample was selected. In contrast, the quality of the other three types of buildings varies. We discussed it with local experts in ancient architecture. Then, we decided on 20 representative samples of these three types of buildings. Richness and evenness [23] exemplify the diversity of lucky aspects the most.

1.  Margalef index

This index reflects the richness of auspicious elements in each building type, which is calculated as follows.

$$D_{ma} = (S - 1)/\ln N \tag{1}$$

S is the number of auspicious elements in each building type, and N is the number of all individuals of auspicious elements in the building type.

2.  Shannon–Wiener index

The Shannon–Wiener index describes the disorder and uncertainty in the occurrence of elements. A better Shannon–Wiener index score indicates a greater diversity of lucky elements. Its calculation formula is as follows.

$$H' = -\sum_{i=1}^{S} P_i \log_2 P_i \tag{2}$$

S is the number of auspicious elements in each building type and $P_i$ is the proportion of elements belonging to cultural category i in the total number of features N.

3.  Simpson's index

Simpson's index, also known as the dominance index, expresses how evenly the individuals in a community (buildings) are distributed among the different species (auspicious elements). The greater the value of Simpson's index, the greater the diversity of lucky elements. Moreover, its calculation formula is as follows.

$$D = 1 - \sum_{i=1}^{S} n_i(n_i - 1)/N(N - 1) \tag{3}$$

N is the number of all individuals of auspicious elements in the building type and $n_i$ is the number of individuals of auspicious part i.

2.2.3. Analysis of the Characteristics of the Co-Occurrence Network of Auspicious Elements

The social network analysis (SNA) method originated from the sociology of measurement. It is also used to study how different people in society interact with each other. The fundamental premise is to construct a "network", using actors as "points" and their relationships as "links" [24]. Auspicious element co-occurrence networks are modelled, calculated, and examined in three steps. First, each auspicious part is considered a "node". There are different lucky elements in building components of the same building type. Co-occurrence describes the relationship between common auspicious elements. If two nodes happen to occur together, it is written as "1", and if they do not, it is written as "0". Next, we made a co-occurrence matrix of lucky elements for each building type. Then, we built each building type's "auspicious element co-occurrence network model" using Ucinet analysis software. Lastly, different metrics are used to look at how a network works, including the network's density, K-core, absolute centrality, intermediate centrality, degree centrality, and intermediate centrality. They are chosen to study each building type's auspicious element co-occurrence network model. Specifically, co-occurrence networks in structural stability and network centrality. The selected indexes are as follows.

1. Network density

Network density is the ratio of the network's actual number of connected relationships to its theoretical maximum. It is applied to measure the overall structural density of the network. As well as the degree of association between network components. It is calculated as follows.

$$P = L/[n(n-1)/2] \tag{4}$$

P is the network density, L is the number of connections, and n is the number of nodes in the network.

2. K-core

The "K-core" is used to analyse the stability of the network's internal structure. Any point in the K-core network structure is adjacent to at least K nodes. The more stable the network topology is, the higher the K-value and the more influential the proportion of K-core.

3. Cut point

It refers to some nodes in a network. If removed, the whole network will be multiple unrelated parts. However, the network's vulnerability can appear by the number and proportion of cut points.

4. Degree centrality potential

The degree centrality potential is used to analyse the centrality of the network. Describe the overall tendency of nodes to congregate at a specific point in the network, and also quantify the network structure's degree of general equilibrium. The calculation formula is as follows.

$$C = \frac{\sum_{i=1}^{n}(C_{max} - C_i)}{max[\sum_{i=1}^{n}(C_{max} - C_i)]} \tag{5}$$

$C_{max}$ is the maximum value of the degree centrality of each node in the network and $C_i$ is the centrality of node i.

5. Intermediary centrality

The network's intermediary nodes are those in the shortest path between other nodes. Intermediary centrality, computed as follows, increases as nodes occur in linked courses.

$$C_{RBi} = \frac{\sum_{j}^{n} \sum_{k}^{n} \frac{g_{jk}(i)}{g_{jk}}}{n^2 - 3n + 2} \tag{6}$$

$C_{RBi}$ is the relative intermediate centrality of nodes, $g_{jk(i)}$ is the number of paths that exist between node j and node k that pass through the third point i, $g_{jk(i)}$ is the number of ways that exist between node j and node k, and n is the total number of nodes.

6. Intermediary centrality potential

It looks at how evenly the intermediary centrality of all the nodes in the network is spread out. The formula for figuring out how to measure this is as follows.

$$C_B = \frac{\sum_{i=1}^{n}(C_{RBmax} - C_{RBi})}{n-1} \tag{7}$$

$C_B$ is the intermediary centrality potential of the network, $C_{RBmax}$ is the maximum possible value of the intermediary centrality of a node, $C_{RBi}$ is the actual value of the intermediary centrality of node i, and n is the number of nodes in the network.

## 3. Results

### 3.1. Cultural Connotation of Auspicious Elements

3.1.1. Expression of Auspicious Culture

We did a study and analysis in the area. The results showed us how Shaxi ancient town's Bai architecture symbolises luck. It takes the building components as the expression carrier. The pursuit of auspicious symbolism is expressed through decorative elements with auspicious meaning. Extensive woodwork, small woodwork, tilework, and colour painting are the principal expression carriers. In addition, there are stone carvings, column bases, and ridge decorations. (See Figure 2).

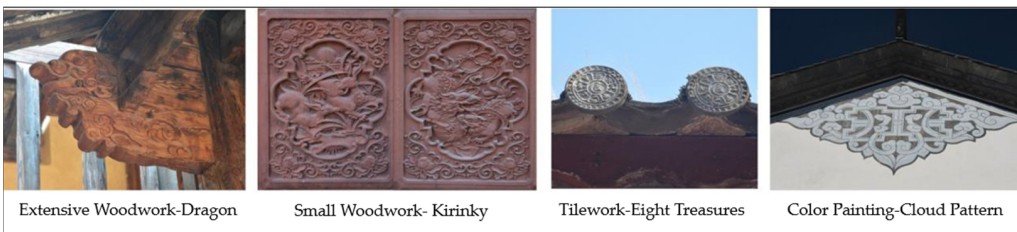

**Figure 2.** Pictures of auspicious elements.

3.1.2. Local Expression of Auspicious Elements in Bai Traditional Architecture

Table 1 shows that the Shaxi Bai people have fully absorbed and borrowed foreign culture. They were then forming a rich combination of auspicious factors. This is a local interpretation of Shaxi Bai's fortunate foreign architecture. Furthermore, it symbolises the lucky culture of traditional Shaxi Bai architecture. It is a fusion and development of multiple cultures and local Bai culture. It shows how accepting the Bai people are of other cultures. At the same time, it shows the inheritance of their own culture.

**Table 1.** The cultural connotation of auspicious elements [25,26].

| Cultural Sources | Number | Auspicious Elements | Local Expression |
|---|---|---|---|
| Buddhist Culture | F1 | Lotus | The lotus is the first of the eight Buddhist treasures of good fortune because the lotus is also known as "lotus" and "and" with the same sound, symbolising good luck and good fortune. The Bai people mostly use the lotus as harmonious auspicious symbolism. |
| | F2 | Magpie | In Buddhism, the magpie represents good fortune and is also called the god of happiness. The Bai people often use the wood carving of "Magpie on the plum tree" or "Joy on the plum tree" on the door to signify the arrival of joyful events. |
| | F3 | Elephant | The white elephant is a Buddhist beast, and Pu Xian, one of the Four Great Bodhisattvas of Buddhism, is a white elephant. |
| | F4 | Curly grass | In the long-term historical development, the scrolling grass pattern is often associated with Buddhist decorative designs. |
| | F5 | Chiwen | The Chiwen, a Buddhist protector, drives away evil, mainly in the form of fish and dragons. |
| | F6 | Pomegranate | The pomegranate was introduced to China on the Silk Road, and the Bai people used it to signify many sons and blessings. |

| Cultural Sources | Number | Auspicious Elements | Local Expression |
|---|---|---|---|
| | F7 | Lion | The lion is the mount and incarnation of Manjushri, and the Bai people take the lion as their primary protector and use the word "lion" to express the meaning of peace and good fortune. |
| | F8 | Eight Treasures | The eight Buddhist treasures are the precious umbrella, the goldfish, the precious vase, the lotus flower, the dharma conch, the auspicious knot, the darling banner, and the Dharma wheel. |
| | F9 | Ruyi | Ruyi is considered the spokesperson of Buddhism, and the Bai people use it to express auspiciousness, health, and longevity. |
| | F10 | Peacock | The peacock is born from the phoenix in Buddhist mythology, and the Peacock King is one of the Tantric masters, which the Bai people use as a symbol of beauty. |
| | F11 | Pisces | The double fish is one of the eight auspicious things in Tibetan Buddhism in China, representing Buddha's two eyes. |
| | F12 | Swastika pattern | The swastika is one of the thirty-two great appearances of the Buddha. |
| Taoist Culture | D1 | Foo (Bat) | The word "Fu" is homophonic with "bat", and the Bai people often use the image of a bat to represent the meaning of blessing and prosperity. |
| | D2 | Locus | "Lu" and "deer" have the same sound, Taoism with the concept of immortality, the deer as a "life of a thousand years" thing, is a symbol of longevity. The Bai people mostly use the "deer and crane with spring" or "six contracts of longevity" pattern to symbolise longevity and good fortune. |
| | D3 | Longevity | The Longevity Star is one of the three stars of Taoism, and the Bai people use longevity to express the meaning of longevity and good fortune. |
| | D4 | Like | Xi is derived from Taoist mythology and is used by the Bai to express joy. |
| | D5 | Crane | The crane is a bright Taoist object, and the Bai people use it to express the meaning of longevity and the prosperity of all things. |
| | D6 | Dragon | The dragon originated from Taoism, and the Bai people use it as a bright object. The Bai use it to express peace, good fortune, and peace in their homes. |
| | D7 | Rabbit | The "jade hare pounding medicine" is a Taoist allusion commonly found on the Baigfan door, signifying longevity. |
| | D8 | Plum (Compound petal pattern) | In Taoism, the plum blossom represents the realm of Yang Sheng, and the Bai's door is often decorated with a combination of plum blossoms and magpies to signify "joyfulness". |
| | D9 | Gourd | The gourd is a representative plant of Taoism. |
| | D10 | Plantain | The plantain is a representative plant of Taoism. |
| | D11 | Peach | The peach is a Taoist immortal fruit used by the Bai to ward off demons on the one hand and signify longevity on the other. |

**Table 1.** *Cont.*

| Cultural Sources | Number | Auspicious Elements | Local Expression |
|---|---|---|---|
| | D12 | The Eight Immortals | The Eight Immortals are the eight Taoist gods and goddesses, and there are famous sayings such as the Eight Immortals crossing the sea. |
| | D13 | Pine | Pine is closely associated with Taoist culture and is known as the immortal of trees, used as a metaphor for longevity and longevity. The Bai people often use it as a symbol of longevity and purity. |
| | D14 | Chrysanthemum (Spoke pattern) | The chrysanthemum is synonymous with "contentment" in Taoist thought. The Bai people use the chrysanthemum pattern as one of the themes for decorating the Bai lattice fan door, mainly for its cold resistance, unchanging quality, and meaning of longevity. |
| Confucianism | R1 | Kirin | The symbol of the Bai people's literary thinking and optimistic attitude towards life is the "Kilin sending papers". |
| | R2 | Phoenix | The phoenix is a Confucian sacred animal. Because of its divine qualities of omen, virtue, metaphorical love, and warding off evil, the Bai see it as a symbol of good fortune, love, and national culture. |
| Bai Culture | B1 | Tile Cat | The tile cat is one of the representatives of the native culture of the Bai people and is used to avoid evil spirits, receive good fortune and calm the house. |
| | B2 | Golden chicken | The chicken is called the "Golden Rooster", as totem worship of the Bai people because the "Gong" of the rooster and the "Gong" homophonic are applied to signify merit and wealth. |
| | B3 | Tiger | Bai people, still white, worship the white tiger. Bai people call themselves "Bai Zi", "Bai Luo Luo", and "Luo" is the white language "tiger". "Luo" means "tiger" in the Bai language. |
| Commercial Horse Culture | C1 | Horse | As an essential ancient marketplace on the old tea horse route, the horse was a necessary means of transportation, and its use as an auspicious ornament meant smooth sailing. |
| Traditional Folk Culture | Z1 | Palindromic pattern | It meant auspicious, deep, and long. The Bai often back to the pattern as a secondary decoration to highlight the main body, forming a contrast effect between the background and the main body. |
| | Z2 | Cloud pattern | As an ancient Chinese auspicious motif, symbolising high ascension and good fortune, the Bai often use cloud patterns together with images of sacred animals and beasts to highlight the carved subject and make painted decorations at a later stage to play a role in praying for blessings. |
| | Z3 | Peony | The peony is an essential subject of traditional Chinese art. The Bai lattice fan doors are often decorated with "phoenix through peony" motifs to signify good fortune, beauty, and prosperity. |
| | Z4 | Bogu | The Bo Gu decoration originates from traditional Chinese culture, and the Bai people use it to express their knowledge of the past and the present and their respect for elegance. |
| | Z5 | Bamboo | Bamboo is a metaphor for a talented gentleman and has the auspicious meaning of "Broadcast peace". |

*3.2. Identification and Quantification of Auspicious Elements*

Table 2 lists and quantifies auspicious decorative elements. There were identified 26 auspicious elements from Xingjiao Temple; 25 auspicious factors in the Patron God Monastery; 21 auspicious aspects from the ancient theatre; 25 auspicious elements from the essential protected dwellings; 26 auspicious elements from the generally covered dwellings; and 23 bright factors from the stores. Different building types use different decorative elements to express lucky culture.

**Table 2.** Extraction table of auspicious elements.

| Building Type | Element Name | Number | Total |
|---|---|---|---|
| Xingjiao Temple | Magpie, Crane, Deer, Rabbit, Elephant, Lion, Dragon, Chiwen, Kirin, Phoenix | 10 | 26 |
| | Bamboo, Pine, Curly Grass, Pomegranate, Chrysanthemum, Plum, Lotus | 7 | |
| | Horse, Pisces, Compound Petal Pattern, Spoke Pattern, Buddhist Eight Treasures, Ruyi, Palindromic Pattern, Cloud Pattern | 8 | |
| | Bogu | 1 | |
| The Patron God Monastery | Magpie, Crane, Deer, Peacock, Kirin, Bat, Lion, Dragon, Elephant, Chiwen, Phoenix | 11 | 25 |
| | Pine, Peach, Plum, Gourd, Curly Grass, Peony, Lotus, Plantain | 8 | |
| | Cloud Pattern, Horse, Longevity, Spoke Pattern, Compound Petal Pattern | 5 | |
| | Bogu | 1 | |
| The ancient theatre | Crane, Tiger, Chiwen, Golden Rooster, Horse, Lion, Phoenix, Dragon, Elephant | 9 | 21 |
| | Pine, Plum, Curly Grass, Chrysanthemum, Lotus | 5 | |
| | Longevity, Spoke Pattern, Compound Petal Pattern, Horse, Palindromic Pattern, Cloud Pattern | 6 | |
| | Eight Immortals | 1 | |
| The essential protected dwellings | Dragon, Peacock, Crane, Phoenix, Horse, Magpie, Rabbit, Kirin, Bat, Deer | 10 | 24 |
| | Plum, Chrysanthemum, Peach, Pine, Peony, Curly Grass, Lotus | 7 | |
| | Like, Cloud Pattern, Spoke Pattern, Horse, Compound Petal Pattern, Palindromic Pattern, Swastika Pattern | 7 | |
| | Bogu | 1 | |
| The generally protected dwellings | Magpie, Peacock, Deer, Rabbit, Phoenix, Dragon, Elephant, Tile Cat, Crane, Bat, Kirin | 11 | 26 |
| | Plum, Chrysanthemum, Curly Grass, Pine, Peony, Lotus | 6 | |
| | Horse, Like, Compound Petal Pattern, Spoke Pattern, Cloud Pattern, Longevity, Palindromic Pattern, Fu | 8 | |
| | Bogu | 1 | |
| Stores | Kirin, Rabbit, Phoenix, Magpie, Crane, Dragon, Deer | 7 | 22 |
| | Peony, Plum, Pomegranate, Pine, Chrysanthemum, Lotus, Curly Grass | 7 | |
| | Horse, Like, Spoke Pattern, Compound Petal Pattern, Cloud Pattern, Palindromic Pattern, Longevity | 7 | |
| | Bogu | 1 | |

### 3.3. Diversity of Auspicious Elements

3.3.1. Diversity of Auspicious Elements of Each Building Type

Table 3 shows that: The Patron God Monastery has the highest Margalef index and Simpson index. At the same time, its Shannon–Wiener index is also a higher value. It means that one type of building out of the six has the most richness and evenness. It also shows that it is the building in the old town with the most diverse lucky elements. Moreover, three diversity indicators of Xingjiao Temple are near the Patron God Monastery. It indicates that the diversity of lucky elements in Xingjiao Temple is high. The three diversity indexes of the ancient theatre are the lowest. It means that it is the building in the old town that has the lowest number of diverse lucky elements because it has only a single performance function.

**Table 3.** Diversity of auspicious elements by building type in Shaxi.

| Building Type | Dma | Ds | H' |
|---|---|---|---|
| Xingjiao Temple (XT) | 7.2135 | 0.9819 | 4.5625 |
| The Patron God Monastery (PGM) | 7.7213 | 0.9877 | 4.5561 |
| The ancient theatre (AT) | 5.9395 | 0.9754 | 4.2542 |
| The essential protected dwellings (EPD) | 6.4691 | 0.9630 | 4.3217 |
| The generally protected dwellings (GPD) | 6.6887 | 0.9663 | 4.4516 |
| Stores (S) | 6.2365 | 0.9803 | 4.3492 |

For ease of representation, the parentheses in the table are shorthand for the sample, such as Xingjiao Temple (XT), similarly after this. And $D_{ma}$ is the Margalef index; $D_s$ is the Simpson index; $H'$ is the Shannon–Wiener index.

Table 3 also shows that the three indexes of the generally protected and the essential protected dwellings are close. It was possible to verify that the two building types are the same because of the fast growth of tourism and urbanisation. Most generally protected homes have been turned into stores and inns for tourists. Therefore, it brings different cultures. It may also explain why the store's three indicators are near the generally protected dwellings.

3.3.2. Diversity of Auspicious Elements of Traditional Architecture in Ancient Towns

Table 3 shows that the Simpson indexes of all six building types in the ancient town are close to 1. This indicates that the historic town's lucky elements of traditional buildings are diverse. It reflects their multicultural background due to history, tourism, and urbanisation.

### 3.4. Analysis of Auspicious Element co-Occurrence Network Characteristics

3.4.1. Overall Network Completeness

Figure 3 shows that the generally protected dwellings and stores are the two building types with a high network density. This indicates that their network structure is complete and they have a high degree of node connectivity. In contrast, the network densities of the other four building types are relatively close. From high to low, they are the ancient theatre, the Xingjiao Temple, the essential protected dwellings, and the Patron God Monastery, in that order. The network density represents the degree of completeness and it shows the closeness of the network structure of auspicious elements. It reflects the degree of auspicious cultural integrity. For example, generally protected dwellings and stores are two typical types of buildings because they are the most influenced by tourism in ancient towns. The high network density indicates their high level of multiculturalism. In addition, the Xingjiao Temple, the Patron God Monastery, and the ancient theatre are representative public buildings in the old town. The government protects them, which leads them to being less influenced by the new foreign culture so their network density is lower.

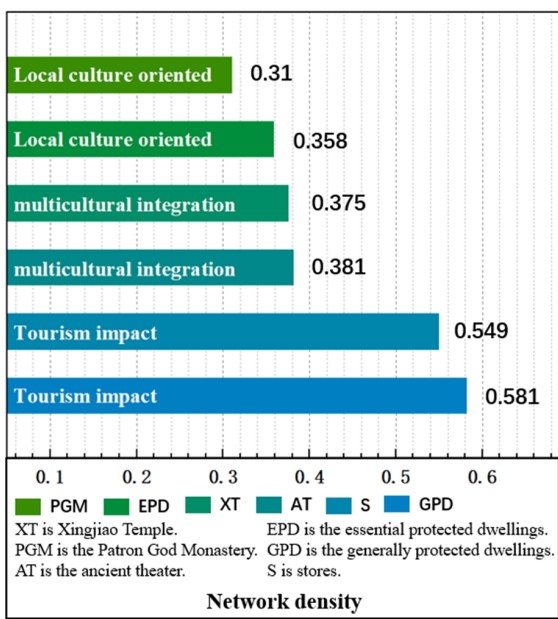

**Figure 3.** Co-occurrence network density bar graph.

3.4.2. Network Local Stability

Figure 4 shows that each type of building in the old town is made up of a group of lucky elements with stable structures and close links. They reflect the parts that have a particular way of being put together to show the lucky culture. Each building type forms a unit cluster family with different cultural meanings. It suggests that the old town's lucky culture comes from a blend of cultures.

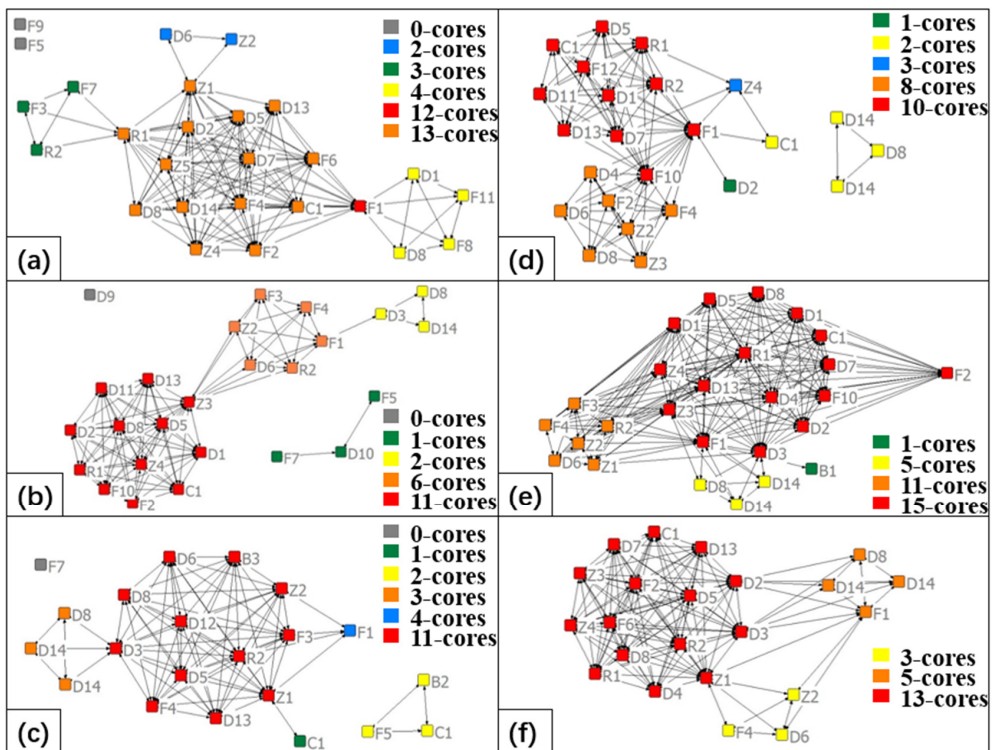

**Figure 4.** K-core diagram of auspicious elements for each building type. (**a**) is Xingjiao Temple, (**b**) is the Patron God Monastery, (**c**) is the ancient theater, (**d**) is the essential protected dwellings, (**e**) is the generally protected dwellings, (**f**) is Stores.

### 3.4.3. Network Vulnerability

Table 4 shows the number of cut points of all architectural types of auspicious elements below 3. The co-occurrence network of lucky architectural elements is relatively safe. Moreover, all of them have formed a stable network structure. In addition, F1 (lotus flower) is the most frequent auspicious element in each building type. It reflects the more significant Buddhist impact on historic town auspicious constructions.

**Table 4.** Co-occurrence network cut points of auspicious elements by building type.

| Building Type | Cutting Point | Number |
| --- | --- | --- |
| Xingjiao Temple (XT) | F1 (Lotus), R1 (Kirin), Z1 (Palindromic pattern) | 3 |
| The Patron God Monastery (PGM) | F1 (Lotus), D3 (Longevity), D10 (Plantain) | 3 |
| The ancient theatre (AT) | D3 (Longevity), Z1 (Palindromic pattern) | 2 |
| The essential protected dwellings (EPD) | F1 (Lotus) | 1 |
| The generally protected dwellings (GPD) | F1 (Lotus) | 1 |
| Stores (S) | None | 0 |

### 3.4.4. Intermediary Centrality

Figure 5 indicates that every building has lucky parts with high intermediate centrality. They are at the heart of the building types' fortunate elements. Each historic town building has a primary and secondary auspicious aspect. Individually, each building type almost always contains the lucky element F1 (lotus). Moreover, it has high intermediate centrality values. The lotus is vital to the ancient town's auspicious architecture. Furthermore, it also reflects Buddhism's relevance in Shaxi's old town's fortunate architecture.

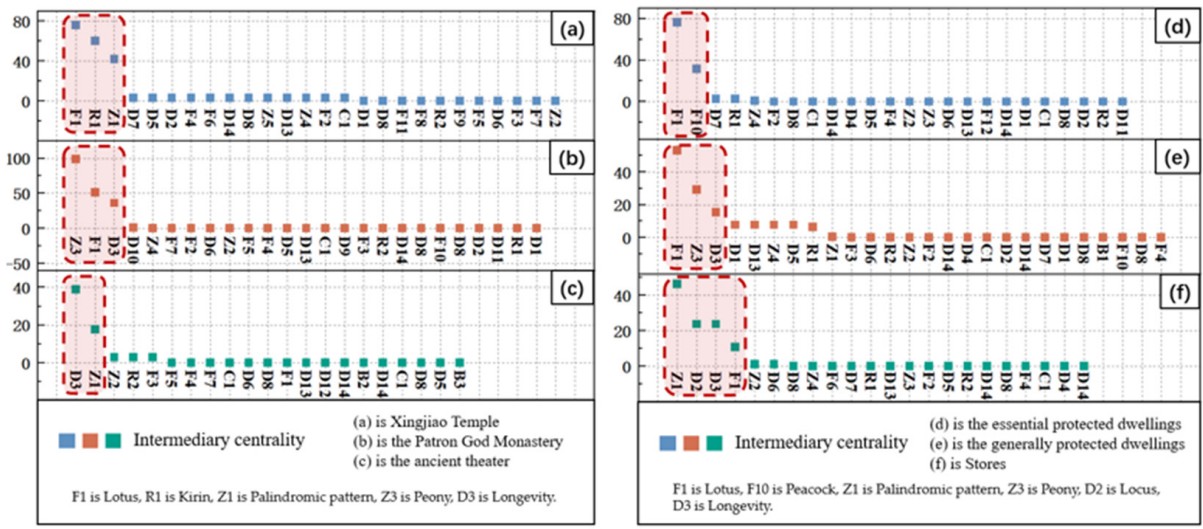

**Figure 5.** Scatterplot of intermediary centrality in descending order.

### 3.4.5. Degree Centre Potential

Figure 6 shows that the degree centrality potential of EPD (55.73%) is significantly higher than the other five building types. It indicates other building types' poorer integrity. However, the network degree centrality of EPD is apparent. It reflects that foreign cultures and better protected with less impact. In addition, most of EPD are residences of historically famous families. The owners have good economic ability and a high cultural level. It leads to the owners having significant autonomy in selecting auspicious elements. Therefore, there is a prominent theme in expressing auspicious culture in EPD. Moreover, the degree centrality potential of Xingjiao Temple (28.67%) is at the lowest level. It indicates that its network centrality is poor and its balance is high.

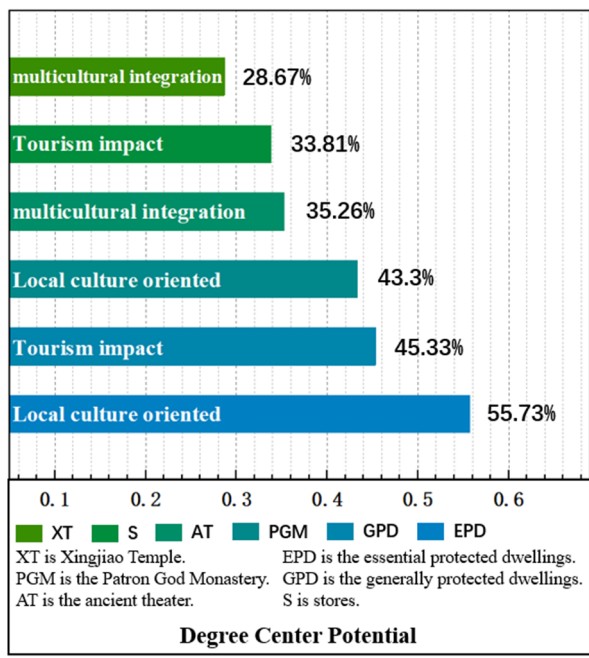

**Figure 6.** Bar chart of degree centre potential.

3.4.6. Intermediary Centre Potential

Figure 7 shows that the PGM's intermediary centrality potential (34.54%) is high. This indicates that its overall intermediary centrality potential is high. It also shows that the overall centrality of its network is increased. This is due to the nature of the temple as a typical building of the Bai's unique Lord worship culture. Therefore, its ability to resist the impact of foreign culture is vital. In contrast, the intermediary centrality potential of GPD (16.63%) is low. It indicates that its overall intermediary centrality is low. Meanwhile, the overall centrality of the network is low. This reflects the more homogeneous distribution of auspicious elements. It reflects the more significant impact of foreign culture brought by tourism development.

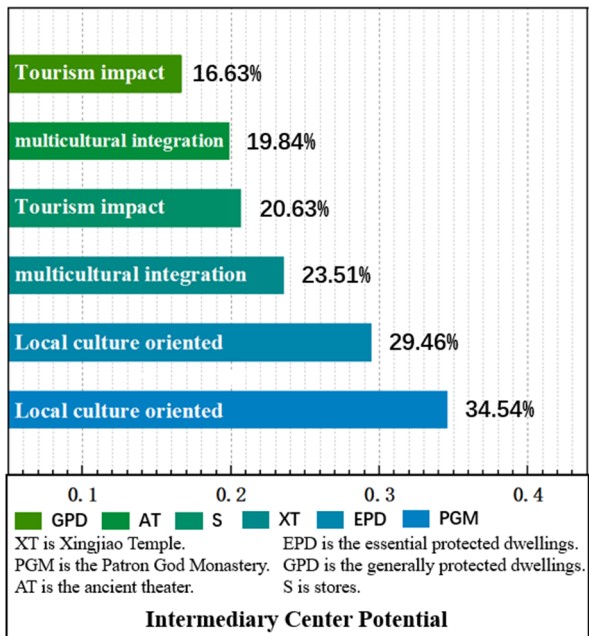

**Figure 7.** Intermediary centre potential bar graph.

## 4. Discussion

### 4.1. Shaxi Bai Traditional Architecture Auspicious Cultural Expression

This study demonstrates that wood, stone, and other decorative arts are employed frequently. The structure has extensive woodwork, small woodwork, bright paintings, and tiles. They are the main thing that people express lucky culture. On the one hand, several research studies on lucky architectural ornamental designs [27–30]. They focused on extensive and small woodwork, ridge creatures, pedestals, and brilliant paintings. However, the tiles are rarely included as an architectural element. On the other hand, much research has investigated the tile pendant's lucky culture [31–33]. Nevertheless, most have concentrated on tile pendants as a distinct architectural component. The study revealed that each building type's lucky elements cluster into unit clusters. Moreover, tile pendants play a particular function in the manifestation of lucky culture. Based on this, the tiles should form a complete system with other elements. It is used to study the lucky details in traditional architecture. It provides a new perspective on building preservation and renovation.

### 4.2. Traditional Village Development Issues Reflected by the Diversity Levels of Lucky Elements

In this study, the diversity levels of each building type's lucky elements are close. It showed that all building types display auspicious features diversely. For example, most of the essential protected dwellings were famous family homes. Their economic and cultural levels were higher than the generally protected dwellings. Therefore, their need to express their lucky culture was higher than in other dwellings. In contrast, the generally protected dwellings are now occupied mainly by ordinary villagers. They should define lucky culture as less than the essential protected dwellings. However, this study indicates that the original gap between them is narrowing. It is because the expansion of tourism has had a significant impact on how people in general live. Most of these dwellings are now being transformed into tourist inns. The essential protected dwellings are learning objects during rehabilitation. The different cultural backgrounds of the owners of tourist inns. Together, they diversify the lucky elements of the two building types. It reflects the multi-cultural integration feature in traditional villages' development process. Furthermore, it illustrates many traditional villages' development issues [34,35]. Traditional settlements must balance different cultures with the original. Perhaps increasing the cultural confidence of local villagers is a solution [36].

### 4.3. Factors Influencing the Expression of Auspicious Culture in Traditional Shaxi Bai Architecture

4.3.1. The Influence of Multiculturalism and Local Culture on the Expression of Bai Auspicious Culture

This study shows that the mediation centrality of the Patron God Monastery is at a high level. Therefore, it is an architectural type with high network centrality. It reflects its high degree of resistance to the impact of foreign culture. It also reflects its thematic expression of auspicious culture. Moreover, this is caused by the strong local culture of lord worship. Some scholars considered Bai lord worship a "cultural confrontation" [37]. It maintains social order and reconstructs local cultural significance. Furthermore, it is compatible with the findings of this study. This study found Xingjiao Temple's auspicious element network less central. It reflects that the lucky aspects are more equal. Furthermore, it demonstrates the diversity of its auspicious cultural expression. Xingjiao Temple is a representative public building. It combines Buddhist Acharya, Taoism, Confucianism, Bai culture, and commercial horse culture. Therefore, mixing cultures creates a more uniform and diverse lucky culture.

Traditional architecture's dominant culture influences auspicious culture. Lucky culture is the dominant culture's outward expression. The Bai people have a custom of worshipping the lord. The temple of the local lord has become an essential part of the Bai villages. Because it is where people go to worship, this is why the native culture has stayed the same. It shows how strong the native culture is and how confident the Bai people are in their culture.

Traditional villages should be passed down and kept safe for as long as possible. Furthermore, both the local culture and the foreign culture should be accepted. Today, many traditional villages must unite people of different cultures, so many villages have become similar, and the area's culture is slowly dying out. However, the old town of Shaxi is a good example. The local culture may depend on the people's confidence in their culture.

### 4.3.2. The Influence of Buddhist Culture on the Expression of Bai Auspicious Culture

This study indicates that the lotus flower is central to Shaxi Bai architecture. Moreover, the lotus flower is the first of the eight Buddhist treasures. Reflecting the significant influence of Buddhism on Bai culture, Dali was known as the "Land of Myriad Fragrances and Buddhism". Many kings of Dali practised Buddhism during the Dali period. The study [38] demonstrates that Buddhism significantly impacted Dali's Bai government. It also spreads culture, education, and societal standards. As a result, Buddhist culture plays a vital role in expressing the auspicious culture. It is the most common element of good luck in the traditional architecture of the Shaxi Bai.

### 4.4. Strategies for the Preservation and Development of the Auspicious Culture of Traditional Shaxi Bai Architecture

The traditional architecture of the Shaxi Bai has its way of showing that things will go well. In terms of public architecture, the Xingjiao Temple is a Buddhist-dominated public building. Moreover, the Patron God Monastery is a public building dominated by the native culture. Due to dominating culture, they have various auspicious elements. At the same time, these buildings are also icons of Shaxi Bai's traditional settlement. Development and changes are prohibited from perpetuating their dominant cultural status. According to researchers, "repair the old as the old, minimising intervention and maximising preservation" [39]. In terms of residential architecture, a unified conservation standard should be issued. The essential protected dwellings have significant historical and cultural value. They should avoid over-growth and blind development. In contrast, the generally protected dwellings should be built sensibly. This way fits within the rules of conservation. It is to meet the social needs of the tourism industry, which is proliferating. Cultural integration is a general trend. It has led to an increase in auspicious cultural diversity. Building local communities' cultural confidence is the essential solution. Furthermore, the Patron God Monastery is a good example.

### 5. Conclusions

Shaxi ancient town is "the only surviving ancient market in the world". Bai culture has been exchanged and absorbed here from ancient times. However, today's rapid tourism development and urbanisation. It has led to a deepening cultural fusion in the town. Traditional architecture is an essential part of traditional villages. Overdevelopment and ethnic and cultural loss challenges are severe threats today. We started with the auspicious culture of Shaxi Bai's traditional architecture. We then analysed the old town's auspicious elements, auspicious cultural expression, and cultural influences. Shaxi Bai's traditional building shows off its lucky cultural traits. The study concludes that Buddhist culture affected Shaxi Bai's traditional architecture. However, the Bai people also keep and spread the local culture. It reflects the coexistence of multiple cultures. Moreover, the Patron God Monastery shows its confidence in the local culture. It is this characteristic that they show when facing cultural integration. The study's results show how the Bai people feel about their culture and how confident they are. It has led to their acceptance of multiculturalism without losing their native culture as well as understanding how to inherit and develop the culture in traditional architecture. The cultural outlook of the Bai people may be a good case to discuss.

**Author Contributions:** Methodology, H.Z.; data curation, H.Z.; writing—original draft preparation, H.Z.; visualization, H.Z.; software, C.D. and Y.R.; validation, Y.R.; investigation, H.Z.; writing—review and editing, Z.H.; supervision, Z.H. All authors have read and agreed to the published version of the manuscript.

**Funding:** This research was funded by the National Nature Science Foundation of China (NSFC) project (Grant numbers 51868008 and 51978187). Funder: National Natural Science Foundation of China.

**Institutional Review Board Statement:** Not applicable.

**Informed Consent Statement:** Informed consent was obtained from all participants in this study.

**Data Availability Statement:** Not applicable.

**Acknowledgments:** The authors are thankful to reviewers and editors for their insightful comments and suggestions to an earlier edition of this article.

**Conflicts of Interest:** The authors declare no conflict of interest.

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
