# Peer review of "The Decorative Auspicious Elements of Traditional Bai Architecture in Shaxi Ancient Town, China"

_sustainability, doi:10.3390/su15031918_

Round 1
Reviewer 1 Report
I would like to congratulate the authors on their detailed and laborious work. While the amount of work is undeniable I would like to point out some ways in which it will be easier for international readers to understand the theme of this research:
1. the abstract does not reflect the methodology
2. the title should be clarified and maybe simplified (I suggest rethinking "their auspicious cultural connotations")
3. the authors MUST introduce earlier in the text an explanation of the notion of "auspicious" and auspicious elements. Also, there is excessive use of the word auspicious in the first part of the manuscript; it can become redundant. I found table 1 very interesting and informative, I think it would serve better if it was included sooner in the presentation
4. the methodological part needs a bit of clarification: "based on the oral experience of local experts and artisans in ancient architecture," - what made them experts? how many? what were they asked? Was there a discussion or an interview? Where were these discussions/interviews conducted? What questions were included? How were the experts/artisans chosen?
5. the methodological part needs a bit of clarification: "By calculating and analyzing relevant indicators" - what makes the use of each indicator relevant to this type of study?
6. the methodological part needs a bit of clarification: "protected dwellings and stores were too much," - how was the too much part decided? (maybe rewrite this part)
7. does the study have a research question? Also please clarify the objectives. The study seems to try and do everything at once.
8. in the "discussion" section you write "The Bai people set up the indigenous cultural architecture and the multicultural fusion architecture separately, and this planning idea solves the problem of inheritance and development well, reflecting the superior planning wisdom of the Bai people and the power of the Bai indigenous culture". How does it solve the problem of inheritance? What is the problem of inheritance? How is one planing power superior to another? Can one culture be superior to another? Because it uses more "auspicious" elements in its architecture? How can one architectural wisdom be superior to another?
9. I would suggest proofreading the text with the help of a native. Some sentences are unclear and can lead to misunderstandings. One single example (and there are many throughout the text) can be "The addition of foreign cultures leads to higher prosperity in their auspicious cultures."
Author Response
Dear Reviewers:
We would like to thank the reviewers’ for giving us constructive suggestions which would help us both in English and in depth to improve the quality of the paper. Here we submit a new version of our manuscript with the title “The diversity of auspicious elements and their auspicious cultural connotations in traditional Bai architecture in Shaxi ancient town, China” which has been modified according to the reviewers’ suggestions. Efforts were also made to correct the mistakes and improve the English of the manuscript. We mark all the changes in blue in the revised manuscript.
The following is a point-to-point response to the reviewers’ comments.
Comment 1:
The abstract does not reflect the methodology
Response 1:
Thank the reviewers for the comment. Your comment motivated us. We have added the methodological content to the revised abstract, see lines 16-19.
Comment 2:
The title should be clarified and maybe simplified (I suggest rethinking "their auspicious cultural connotations")
Response 2:
Thank the reviewer for the comments, the original title does seem vague and does not reflect the content of the article, we have simplified the title as you suggested, see lines 2-3.
Comment 3:
The authors MUST introduce earlier in the text an explanation of the notion of "auspicious" and auspicious elements. Also, there is excessive use of the word auspicious in the first part of the manuscript; it can become redundant. I found table 1 very interesting and informative, I think it would serve better if it was included sooner in the presentation
Response 3:
Thank the reviewer for the comments, we have made the following changes to the manuscript based on your suggestion.
We have added an explanation of "lucky" and “lucky elements” to the manuscript, see lines 38-43.
I would like to apologize for the excessive use of the word "auspicious" in the first half of the manuscript, which was caused by the fact that English is not my native language, and we have adjusted it by deleting and replacing some of the words, see Section “1. Introduction”.
We agree with you that table 1 is informative. It is a collection of lucky elements from different cultures. But it is ours because we read about them and talked to people in the area. It shows the results of our research. So we planned it was suitable to leave it where it was.
Comment 4:
The methodological part needs a bit of clarification: "based on the oral experience of local experts and artisans in ancient architecture," - what made them experts? how many? what were they asked? Was there a discussion or an interview? Where were these discussions/interviews conducted? What questions were included? How were the experts/artisans chosen?
Response 4:
Thank the reviewer for the comments, which means a lot to our manuscript. We believe that the issues you mentioned are exactly what we missed in writing the manuscript and we will add to them, as described in lines 104-108.
Comment 5:
The methodological part needs a bit of clarification: "By calculating and analyzing relevant indicators" - what makes the use of each indicator relevant to this type of study?
Response 5:
Thank the reviewer for the comments. We have clarified this section, see lines 108-112.
Comment 6:
The methodological part needs a bit of clarification: "protected dwellings and stores were too much," - how was the too much part decided? (maybe rewrite this part).
Response 6:
Thank the reviewer for the comments. This was a misunderstanding caused by an error in our writing. We have revised it, see lines 119-121.
Comment 7:
Does the study have a research question? Also please clarify the objectives. The study seems to try and do everything at once.
Response 7:
Thank the reviewer for the comments, we believe this was due to errors in the preparation of our first draft. And our research questions have been added to the methodological part, see lines 112-114.
Comment 8:
In the "discussion" section you write "The Bai people set up the indigenous cultural architecture and the multicultural fusion architecture separately, and this planning idea solves the problem of inheritance and development well, reflecting the superior planning wisdom of the Bai people and the power of the Bai indigenous culture". How does it solve the problem of inheritance? What is the problem of inheritance? How is one planing power superior to another? Can one culture be superior to another? Because it uses more "auspicious" elements in its architecture? How can one architectural wisdom be superior to another?
Response 8:
Thank the reviewer for the comments. We have revised this part of the discussion based on the questions you asked, see lines 377-386.
Comment 9:
I would suggest proofreading the text with the help of a native. Some sentences are unclear and can lead to misunderstandings. One single example (and there are many throughout the text) can be "The addition of foreign cultures leads to higher prosperity in their auspicious cultures."
Response 9:
Thank the reviewers for the comments. We think this is very important advice because English is our second language. And it's easy for our English to be messed up because of that. We rewrote the whole thing and changed many sentences that weren't clear. Such as line 264.
We have the improper parts revised accordingly. The English writing in this manuscript has been promoted to eliminate the mistakes of grammar and syntax, and the description of the work has been refined to enhance the clarity.
We tried our best to improve the manuscript and made some changes in the manuscript. The quality and clarity of the manuscript have been improved a lot according to your specific and valuable comments. We appreciate for Reviewers' warm work earnestly and hope that the correction will meet with approval. If there are still questions in the correction, we would be delighted to receive your comments.
Once again, thank you very much for your comments and suggestions.
Best wishes,
Zongsheng Huang

Reviewer 2 Report
The manuscript had the potential to be of interest to your readers but unfortunately has serious flaws. There are some incorrect information and misinterpreted results. In many instances, the language is difficult to read and to understand. In the attached file i included more specific comments and some suggestions that may be helpful (recorded using yellow highlight and comments in red).

Author Response
Dear Reviewers:
We would like to thank the reviewers’ for giving us constructive suggestions which would help us both in English and in depth to improve the quality of the paper. Here we submit a new version of our manuscript with the title “The diversity of auspicious elements and their auspicious cultural connotations in traditional Bai architecture in Shaxi ancient town, China” which has been modified according to the reviewers’ suggestions. Efforts were also made to correct the mistakes and improve the English of the manuscript. We mark all the changes in blue in the revised manuscript.
The following is a point-to-point response to the reviewers’ comments.
Comment 1:
Species richness index, measure fot the total number of different species in a given area/community. N = population = the total number of individuals in the sample and S = the number of species recorded. No abundance is taken in consideration.
Response 1:
Thank the reviewers for the comment. This was the result of an error in our writing. And we have revised this section, see line 124. We also have revised the ‘Margrave index’, see line 123.
Comment 2:
Simpson’s Index
1) Formula as presented __ D ranges between 0 (no diversity) and 1 (infinite diversity)
2) Formula without the "1-" __ D ranges between 1 (no diversity) and 0 (infinite diversity)
Simpson's index may be defined in different ways, but the original and simplest is that it is the probability that two individuals drawn at random from an assemblage will belong to the same species. As such it is a measure of dominance, and for a highly dominated (i.e., highly uneven) assemblage the probability of drawing two individuals from the same species will be high (approaching 1). For a completely even assemblage, in which all individuals belong to different species, the probability of drawing two individuals from the same species will be 0. Conventionally, more even assemblages are considered to be more diverse; therefore, this scaling appears counterintuitive as high values imply low diversity. The index is often, therefore, converted from a dominance measure into an evenness (or equitability) measure either by subtracting the dominance value from 1 (Somerfield et al., 2008)
Response 2:
Thank the reviewer for the comments. Your suggestion is crucial, we have a clearer understanding of the Simpson index. This was the result of an error in our writing. And we have revised this formula, see line 138-140. (Simpson's formula is , We switched the position of D.)
Comment 3:
Like Simpson's is an evenness index (redundant).
If all species are represented in equal numbers in the sample, then J’=1. If one species strongly dominates J’is close to zero.
Expresses how evenly the individuals in a community (building type) are distributed among the different species (auspicious elements)
Response 3:
Thank the reviewer for the comments. We believe that your suggestion is very objective and we have removed the Uniformity Index as an indicator.
Comment 4:
The results analysis makes no sense. Indexes were misinterpreted and key conclusions are incorrect and farfetched.
The overall conclusion in the light os this results is that all buildings have a high diversity of auspicious elements, with the PGM being the one with the greatest diversity, and there is no dominance of any element in any of the buildings.
Response 4:
Thank the reviewer for the comments. This was indeed an error caused by our misunderstanding of the Simpson index, and we have revised the results of this section, see lines 227-235.
Comment 5:
Did you mean Shannon index? I wuould like to see a clearer understanding about whats is a aroma index, since this is the first time that is mencionated.
Response 5:
Thank the reviewer for the comments. We apologize that this was the result of a writing error on our part. We have completed the revision, see 245.
Comment 6:
In fact, there is no significant differences. Again, this are farfetched conclusions.
Response 6:
Thank the reviewer for the comments. We believe the issues you raised are objective and we have revised them, see lines 236-242 and lines 244-360.
Comment 7:
This chart (figure 3) provides exactly the same information as Table 3. It is redundant.
Response 7:
Thank the reviewer for the comments. we have removed Figure 3.
Comment 8:
3.4.1 section = 3.4.2 section title
Response 8:
Thank the reviewer for the comments, we have revised this section, see Line 274.
Comment 9:
The network analysis and its conclusions need to be thoroughly reviewed, key findings are incorrect and/or farfetched. Table 4 and figure 6 are only complementary information to figure 5, they do not provide additional information. Maybe you should consider providing a network where all studied places (buildings) are simultaneously included, so that similarities, differences, centralitys, etc, between them can be analysed.
Response 9:
Thank the reviewer for the comments. We believe that your suggestion is objective and correct. Table 4 and figure 6 are complementary information to figure 5. But we have included these two sections in order to provide a more visual representation of our findings. We have revised it, see lines 285-291 and lines 294-300.
We think you are good for building a network of auspicious elements for the whole ancient town. We have built this network model before, but since the auspicious elements in each building type are basically close to each other, it is difficult to get useful results, so we did not add this part. We think maybe this network is used to compare with other traditional villages to get good results.
Comment 10:
4.2 section = 4.4 section title.
No conservation strategy is actually developed and proposed as stated in the research goals of section 1. Introduction.
Response 10:
Thank the reviewer for the comments. We have revised this section, see Line 344. And we have carefully considered that the study of diversity indicators only reflects the current state of multicultural integration in villages today. It does not provide a sufficient basis for proposing conservation strategies for them. So we have rewritten this section, see lines 345-360.
Comment 11:
No supplementary material was provided or mentioned in the text. Remove?
Response 11:
Thank the reviewer for the comments. We have statistics on the auspicious elements and matrix data of the co-occurrence network that will be presented.
We have the improper parts revised accordingly. The English writing in this manuscript has been promoted to eliminate the mistakes of grammar and syntax, and the description of the work has been refined to enhance the clarity.
We tried our best to improve the manuscript and made some changes in the manuscript. The quality and clarity of the manuscript have been improved a lot according to your specific and valuable comments. We appreciate for Reviewers' warm work earnestly and hope that the correction will meet with approval. If there are still questions in the correction, we would be delighted to receive your comments.
Once again, thank you very much for your comments and suggestions.
Best wishes,
Zongsheng Huang

Reviewer 3 Report
This article on the multiculturally auspicious elements of the traditional Bai architecture in the Shaxi Ancient Town in Yunan has produced some interesting findings. The introduction of the research background covered sufficient information and served well for the following analysis. It was necessary to understand the aims and significance of the research, which focused on employing the auspicious symbols in traditional architecture that reflect integrations of diverse cultures.
The combination of qualitative and quantitative methods has effectively added value to the analysis. The research methodology outlined a range of appropriate methods to identify the elements, calculate diversity, and establish the co-occurrence network model. The description of how the data were sampled and compared through the indexes was clear, and the coding framework was convincing. The sections on results presenting and data analysing were coherently linked to the research aims and well-formed using illustrative charts and graphs. The conclusion summarised the findings in relation to the research questions and showed a good awareness of cultural conservation for traditional architecture.
Improvements are as follows:
- There are some vague statements, such as the aim of the study in the Abstract (Lines 15-16) that needs illustrative examples, and the unclear terms of “traditional areas” (Line 49) and the meaning of “she” in the section 3.4.3 (Line 307).
- The article is generally referenced, but some key references are needed, including the introduction of the research background (Lines 86-102), the source of Figure 1, and the definitions of the indexes and analysis methods.
- Repeated subtitles -- 3.4.1 (Line 270) and 3.4.2 (Line 292); 4.2 (Lines 368-369) and 4.4 (Lines 424-425)
- There are some minor grammatical errors – an improper noun of “studies” (Line 56), which should be ‘researchers’ or ‘scholars’; inappropriate punctuations (i.e., Lines 49-52); and an incomplete sentence (Line 319).
- Also, some layout issues – inconsecutive order numbers for the sections (i.e., sequence of numbers 1- 4 under the subsequence of 2.2.2); and the font (Lines 191, 198, 205) are inconsistent.
Author Response
Dear Reviewers:
We would like to thank the reviewers’ for giving us constructive suggestions which would help us both in English and in depth to improve the quality of the paper. Here we submit a new version of our manuscript with the title “The diversity of auspicious elements and their auspicious cultural connotations in traditional Bai architecture in Shaxi ancient town, China” which has been modified according to the reviewers’ suggestions. Efforts were also made to correct the mistakes and improve the English of the manuscript. We mark all the changes in blue in the revised manuscript.
The following is a point-to-point response to the reviewers’ comments.
Comment 1:
This article on the multiculturally auspicious elements of the traditional Bai architecture in the Shaxi Ancient Town in Yunan has produced some interesting findings. The introduction of the research background covered sufficient information and served well for the following analysis. It was necessary to understand the aims and significance of the research, which focused on employing the auspicious symbols in traditional architecture that reflect integrations of diverse cultures.
The combination of qualitative and quantitative methods has effectively added value to the analysis. The research methodology outlined a range of appropriate methods to identify the elements, calculate diversity, and establish the co-occurrence network model. The description of how the data were sampled and compared through the indexes was clear, and the coding framework was convincing. The sections on results presenting and data analysing were coherently linked to the research aims and well-formed using illustrative charts and graphs. The conclusion summarised the findings in relation to the research questions and showed a good awareness of cultural conservation for traditional architecture.
Response 1:
Thank the reviewers for the comment.
Comment 2:
There are some vague statements, such as the aim of the study in the Abstract (Lines 15-16) that needs illustrative examples, and the unclear terms of “traditional areas” (Line 49) and the meaning of “she” in the section 3.4.3 (Line 307).
Response 2:
Thank the reviewer for the comments. We think your suggestion is very critical, and we have made the following changes in accordance with your suggestion.
We have revised the abstract. It adds a description of the subject, see lines 14-16.
We have revised the introduction. 'traditional areas' revised to 'traditional villages', see line 45.
We have revised ‘she’, see line 288.
Comment 3:
The article is generally referenced, but some key references are needed, including the introduction of the research background (Lines 86-102), the source of Figure 1, and the definitions of the indexes and analysis methods.
Response 3:
Thank the reviewer for the comments. We have made the following changes to the manuscript based on your suggestion.
We have added references for the study area, see line85, line 87, line 89, line 90.
We revised Figure 1 with an explanation of his sources, see 93-94.
We added references to the analysis methods, which are used to illustrate the feasibility of the analysis method, see line 108-114.
Comment 4:
Repeated subtitles -- 3.4.1 (Line 270) and 3.4.2 (Line 292); 4.2 (Lines 368-369) and 4.4 (Lines 424-425)
Response 4:
Thank the reviewer for the comments, we have revised the repeated subtitles, see 3.4.2 (Line 274) and 4.2 (Lines 344).
Comment 5:
There are some minor grammatical errors – an improper noun of “studies” (Line 56), which should be ‘researchers’ or ‘scholars’; inappropriate punctuations (i.e., Lines 49-52); and an incomplete sentence (Line 319).
Response 5:
Thank the reviewer for the comments, we have revised the grammatical errors you raised, see Line 52; Line 45-47; Line 298-300.
Comment 6:
Also, some layout issues – inconsecutive order numbers for the sections (i.e., sequence of numbers 1- 4 under the subsequence of 2.2.2); and the font (Lines 191, 198, 205) are inconsistent.
Response 6:
Thank the reviewer for the comments. We checked the sequence numbers and made changes, see line 123, line 128 and line 136. We have revised the Inconsistent fonts, see Line 176, Line 183, Line 190.
We have the improper parts revised accordingly. The English writing in this manuscript has been promoted to eliminate the mistakes of grammar and syntax, and the description of the work has been refined to enhance the clarity.
We tried our best to improve the manuscript and made some changes in the manuscript. The quality and clarity of the manuscript have been improved a lot according to your specific and valuable comments. We appreciate for Reviewers' warm work earnestly and hope that the correction will meet with approval. If there are still questions in the correction, we would be delighted to receive your comments.
Once again, thank you very much for your comments and suggestions.
Best wishes,
Zongsheng Huang

Reviewer 4 Report
Dear Authors,
Thank you for the opportuntiy to review your work. These are the corrections and areas of improvement that you may want to consider:
Line 35: "included" instead of "selected"
Line 130: Is it "Margrave" or "Margalef"? I have a feeling it is the latter
Line 252: Shannon-Wiener seems to be the correct index and not "aroma"
Line 254: Interestingly, "fragrance" seems to be related to "aroma" in 252
Line 319: "in the This"
There is a pressing need to give the reader a preview of the different cultures that are featured in Shaxi Ancient Town. Perhaps it is also helpful that we juxtapose the town with other towns in China that are not exactly flexible in terms of, shall we say, cultural borrowings and integrations.
I would also like to ask you to consider adding material to support your argument regarding tiles being primary expression carriers while scholars appear to overlook these.
I am unsure if "auspicious" is the appropriate word to use for the paper. My understanding is that it is defined as "conducive to success" or "favorable." Is this what you are truly trying to express or is there a better word for it perhaps? I entertain the possibility of a loss in translation as I assume that your first language is not English (it is my second, by the way, which suggests that we are in the same boat). While auspicious culture may be crystal clear to the Chinese ethos, it may not be so for at least some of our readers.
Finally, I request that you retain the legend you used in Figure 3 across all other figures where this will be helpful, Figure 4 being one of them. Where Figure 6 is concerned, I suggest we include the names of the objects/artifacts that the letter-number codes refer to. If there are consistencies across the elements, these can be highlighted as well.
My hope is that you take these suggestions in the spirit with which they were given, that is, with the hope of improving your paper. Thank you and I look forward to receiving a revised version.
Author Response
Dear Reviewers:
We would like to thank the reviewers’ for giving us constructive suggestions which would help us both in English and in depth to improve the quality of the paper. Here we submit a new version of our manuscript with the title “The diversity of auspicious elements and their auspicious cultural connotations in traditional Bai architecture in Shaxi ancient town, China” which has been modified according to the reviewers’ suggestions. Efforts were also made to correct the mistakes and improve the English of the manuscript. We mark all the changes in blue in the revised manuscript.
The following is a point-to-point response to the reviewers’ comments.
Comment 1:
Thank you for the opportuntiy to review your work. These are the corrections and areas of improvement that you may want to consider:
Line 35: "included" instead of "selected"
Line 130: Is it "Margrave" or "Margalef"? I have a feeling it is the latter
Line 252: Shannon-Wiener seems to be the correct index and not "aroma"
Line 254: Interestingly, "fragrance" seems to be related to "aroma" in 252
Line 319: "in the This"
Response 1:
Thank the reviewers for the comment. We think your suggestions are very important. We have made the following changes to the manuscript based on your suggestions.
We have revised the section that introduces, see line 29.
We have revised the error, see line 123.
We have revised the error, see line 245.
We have rewritten this section, see line 230-231.
We have rewritten this section, see line 298-299.
Comment 2:
There is a pressing need to give the reader a preview of the different cultures that are featured in Shaxi Ancient Town. Perhaps it is also helpful that we juxtapose the town with other towns in China that are not exactly flexible in terms of, shall we say, cultural borrowings and integrations.
Response 2:
Thank the reviewer for the comments. We think your suggestion is very critical, and we have add this section , see lines 358-361.
Comment 3:
I would also like to ask you to consider adding material to support your argument regarding tiles being primary expression carriers while scholars appear to overlook these.
Response 3:
Thank the reviewer for the comments. This was the result of a writing error on our part, which we have revised, see lines 334-339.
Comment 4:
I am unsure if "auspicious" is the appropriate word to use for the paper. My understanding is that it is defined as "conducive to success" or "favorable." Is this what you are truly trying to express or is there a better word for it perhaps? I entertain the possibility of a loss in translation as I assume that your first language is not English (it is my second, by the way, which suggests that we are in the same boat). While auspicious culture may be crystal clear to the Chinese ethos, it may not be so for at least some of our readers.
Response 4:
Thank the reviewer for the comments, we think your suggestions are very important. The auspicious in this article is a sign of good luck and expresses the people's view of a brighter future. So we revised the auspicious to lucky, see the title (line 2-3).
Comment 5:
Finally, I request that you retain the legend you used in Figure 3 across all other figures where this will be helpful, Figure 4 being one of them. Where Figure 6 is concerned, I suggest we include the names of the objects/artifacts that the letter-number codes refer to. If there are consistencies across the elements, these can be highlighted as well.
Response 5:
Thank the reviewer for the comments. We have revised the legend in the image you mentioned, see line 272, line 314, line 327.
We have revised Figure 6. The consistency between them we illustrate later, see lines 301.
We have the improper parts revised accordingly. The English writing in this manuscript has been promoted to eliminate the mistakes of grammar and syntax, and the description of the work has been refined to enhance the clarity.
We tried our best to improve the manuscript and made some changes in the manuscript. The quality and clarity of the manuscript have been improved a lot according to your specific and valuable comments. We appreciate for Reviewers' warm work earnestly and hope that the correction will meet with approval. If there are still questions in the correction, we would be delighted to receive your comments.
Once again, thank you very much for your comments and suggestions.
Best wishes,
Zongsheng Huang

Round 2
Reviewer 1 Report
The manuscript was improved; I would suggest further work on improving the English of the manuscript.
It is not yet clear the reason why the authors embarked on this research (other than just cataloguing these elements). Is there a research question or a hypothesis?
The references list is not "international".
Author Response
Dear Reviewers:
We would like to thank the reviewers’ for giving us constructive suggestions which would help us both in English and in depth to improve the quality of the paper. Here we submit a new version of our manuscript with the title “The lucky cultural characteristics of traditional Bai architecture in Shaxi ancient town, China” which has been modified according to the reviewers’ suggestions. Efforts were also made to correct the mistakes and improve the English of the manuscript. We mark all the changes in blue in the revised manuscript.
The following is a point-to-point response to the reviewers’ comments.
Comment 1:
The manuscript was improved; I would suggest further work on improving the English of the manuscript.
Response 1:
Thank the reviewers for the comment. Your comment motivated us. We have improved the English of the manuscript.
Comment 2:
It is not yet clear the reason why the authors embarked on this research (other than just cataloguing these elements). Is there a research question or a hypothesis?
Response 2:
Thank the reviewer for the comments. We have added this section to the abstract, see lines 13-16.
Comment 3:
The references list is not "international".
Response 3:
Thank the reviewer for the comments, we have revised the references list, see lines 454-533.
We have the improper parts revised accordingly. The English writing in this manuscript has been promoted to eliminate the mistakes of grammar and syntax, and the description of the work has been refined to enhance the clarity.
We tried our best to improve the manuscript and made some changes in the manuscript. The quality and clarity of the manuscript have been improved a lot according to your specific and valuable comments. We appreciate for Reviewers' warm work earnestly and hope that the correction will meet with approval. If there are still questions in the correction, we would be delighted to receive your comments.
Once again, thank you very much for your comments and suggestions.
Best wishes,
Zongsheng Huang

Reviewer 2 Report
The authors have corrected previous errors and made some of the proposed changes. Even so, there are small typos throughout the manuscript, such as the numbering of images in the text, lack of spacing, etc (recorded using yellow highlight in the attached file). The language and grammar still needs a extensive revision as in many instances it is difficult to read and understand, which detracts from the overall quality of the article. I’ve also included some suggestions that may be helpful (in blue).

Author Response
Dear Reviewers:
We would like to thank the reviewers’ for giving us constructive suggestions which would help us both in English and in depth to improve the quality of the paper. Here we submit a new version of our manuscript with the title “The lucky cultural characteristics of traditional Bai architecture in Shaxi ancient town, China” which has been modified according to the reviewers’ suggestions. Efforts were also made to correct the mistakes and improve the English of the manuscript. We mark all the changes in blue in the revised manuscript.
The following is a point-to-point response to the reviewers’ comments.
Comment 1:
The authors have corrected previous errors and made some of the proposed changes. Even so, there are small typos throughout the manuscript, such as the numbering of images in the text, lack of spacing, etc (recorded using yellow highlight in the attached file). The language and grammar still needs a extensive revision as in many instances it is difficult to read and understand, which detracts from the overall quality of the article. I’ve also included some suggestions that may be helpful (in blue).
Response 1:
Thank the reviewers for the comment. Your comment motivated us.
We have revised the number of images, see line 258, line 274, line 291, line 301 and line 314.
We have revised the lack of spacing, see line 43 and line 96.
We have revised another grammatical errors, see line 85 and line 219.
We have revised the title, see line 2.
We have revised the instances it is difficult to read and understand, see lines 126-127, lines 139-141, lines 221-222, lines 242-244, lines 246-247 and lines 343-344.
We have the improper parts revised accordingly. The English writing in this manuscript has been promoted to eliminate the mistakes of grammar and syntax, and the description of the work has been refined to enhance the clarity.
We tried our best to improve the manuscript and made some changes in the manuscript. The quality and clarity of the manuscript have been improved a lot according to your specific and valuable comments. We appreciate for Reviewers' warm work earnestly and hope that the correction will meet with approval. If there are still questions in the correction, we would be delighted to receive your comments.
Once again, thank you very much for your comments and suggestions.
Best wishes,
Zongsheng Huang

Reviewer 4 Report
Dear Authors,
Thank you for kindly considering my suggestions.
Author Response
Dear Reviewers:
We would like to thank the reviewers’ for giving us constructive suggestions which would help us both in English and in depth to improve the quality of the paper. Here we submit a new version of our manuscript with the title “The lucky cultural characteristics of traditional Bai architecture in Shaxi ancient town, China” which has been modified according to the reviewers’ suggestions. Efforts were also made to correct the mistakes and improve the English of the manuscript. We mark all the changes in blue in the revised manuscript.
The following is a point-to-point response to the reviewers’ comments.
Comment 1:
Dear Authors,
Thank you for kindly considering my suggestions.
Response 1:
We appreciate your suggestions for us, which are very objective and useful. We sincerely thank you for the help you have provided to us and wish you well in your work and good health.
Once again, thank you very much for your comments and suggestions.
Best wishes,
Zongsheng Huang
